# Risk Factors Association with Transcriptional Activity of Metalloproteinase 9 (MMP-9) and Tissue Inhibitor of Metalloproteinases 1 (TIMP-1) Genes in Patients with Heart Failure

**DOI:** 10.3390/biomedicines12030601

**Published:** 2024-03-07

**Authors:** Józefa Dąbek, Dariusz Korzeń, Oskar Sierka, Lech Paluszkiewicz, Hendrik Milting, Zbigniew Gąsior

**Affiliations:** 1Department of Cardiology, Faculty of Health Sciences in Katowice, Medical University of Silesia in Katowice, Ziołowa Street 45/47, 40-635 Katowice, Poland; 2Voivodeship Specialist Hospital Megrez Sp. z o. o., Edukacji Street 102, 43-100 Tychy, Poland; 3College of Doctoral School, Faculty of Medical Sciences in Katowice, Medical University of Silesia in Katowice, Medyków Street 18, 40-754 Katowice, Poland; 4Clinic of Thoracic and Cardiovascular Surgery, Heart and Diabetes Center North-Rhine Westphalia, University Hospital of the Ruhr-University Bochum, Georg Street 11, 32545 Bad Oeynhausen, Germany; 5Erich and Hanna Klessmann Institute for Cardiovascular Research and Development, Heart and Diabetes Center North-Rhine Westphalia, University Hospital of the Ruhr-University Bochum, Georg Street 11, 32545 Bad Oeynhausen, Germany

**Keywords:** genes, risk factors, heart failure

## Abstract

The aim of the study was to assess the occurrence of classic risk factors in the study group of patients with heart failure and to link them with the transcriptional activity of the examined genes: *metalloproteinase 9 (MMP-9)* and the *tissue inhibitor of metalloproteinases 1 (TIMP-1)*. A total of 150 (100%) patients qualified for the study, including 80 (53.33%) patients with heart failure in the course of coronary artery disease, 40 (26.67%) with coronary artery disease without heart failure, and 30 (20.00%) in whom the presence of atherosclerotic changes in the coronary arteries was excluded. The material for molecular tests was peripheral blood collected from patients within the first 24 h of hospitalisation. A quantitative analysis of transcriptional activity was performed using the RT-qPCR technique. The most common classic risk factors among the patients in the study group were arterial hypertension (117; 78.00%) and overweight/obesity (102; 68%). In the group of patients with coronary artery disease and heart failure burdened with overweight/obesity, a significantly higher transcriptional activity of the *metalloproteinase 9 (MMP-9)* gene was found in comparison to patients who were not burdened with this risk factor. The analysis also showed the statistically significant higher transcriptional activity of the *metalloproteinase 9 (MMP-9)* gene in a group of patients with coronary artery disease and heart failure who smoked. The examined patients with heart failure due to myocardial ischemia were burdened with numerous cardiovascular risk factors, the most common of which were arterial hypertension, obesity/overweight, and hypercholesterolemia. A significant increase in the transcriptional activity of the *metalloproteinase 9 (MMP-9)* gene in the presence of risk factors (male sex, overweight/obesity, smoking) indicates another pathomechanism of their action and participation in the development and progression of heart failure during myocardial ischemia. There is a need for systematic information and educational activities promoting a healthy lifestyle with the elimination of modifiable risk factors for cardiovascular diseases.

## 1. Introduction

Heart failure (HF) is a clinical syndrome with symptoms and/or signs caused by a structural and/or functional abnormality of the heart, confirmed by elevated levels of natriuretic peptides and/or objective evidence of congestion in central and/or peripheral circulation [1].

The most common cause of heart failure is coronary artery disease, caused by atherosclerosis in the coronary arteries, accounting for 98% of the causes of myocardial ischemia. The result of the formation of stenoses is the disturbance of blood flow in the microcirculation and, consequently, damage to cardiomyocytes due to restrictions in the supply of oxygen and nutrients that are necessary for metabolic processes [2]. Additional causes of ischemic heart disease (2%) include vascular diseases in the course of other conditions (e.g., systemic lupus and scleroderma), emboli (e.g., in the course of endocarditis or myxoma), anatomical defects (e.g., aneurysms and arteriovenous fistulas), and others (e.g., anaemia, arrhythmias, and hyperthyroidism).

Various factors and behaviours of a given person contribute to the development of ischemic heart disease based on atherosclerosis. Broadly defined risk factors are traits, states, or behaviours that increase the likelihood of developing a given disease entity or a whole group of diseases [3]. The occurrence of a given factor correlates with the accelerated development of a given disease. These factors can be divided into behavioural (e.g., poor diet), physiological (e.g., high blood pressure), demographic (e.g., gender), environmental (e.g., air quality), and genetic (e.g., having certain genes or genetic mutations) factors. Another division categorises risk factors into the modifiable and the non-modifiable [4]. So far, about 300 risk factors associated with the development of coronary atherosclerosis have been identified [5]. Among them, there are also factors related to heredity. Several polymorphisms affecting cardiovascular risk have been identified. These include, among others, variants of apolipoprotein E (Apo-E), plasminogen activator inhibitor 1 (PAI-1), and angiotensin-converting enzyme inhibitor (ACEI) [6,7]. The impact of risk factors on the molecular basis of cardiovascular disease has not been fully explored; therefore, the research conducted in this area can be considered innovative. Especially interesting is their influence on metalloproteinases and their tissue inhibitors, which play an important role in the remodelling of heart tissue in the progress of described diseases. Among the mentioned enzymes, metalloproteinase 9 (MMP-9) plays a significant role in neovascularisation through the proteolytic degradation of proteins in the basal lamina of blood vessels and the release of a biologically active form of vascular endothelial growth factor [8]. The level of MMP-9 increases rapidly in cardiovascular diseases such as arterial hypertension, atherosclerosis, and myocardial infarction, and a significant number of publications on MMP-9 emphasise its importance as a useful diagnostic and prognostic marker [9]. It was also found that in patients with coronary artery disease, higher MMP-9 levels are an independent risk factor for cardiovascular mortality [10].

In patients with chronic HF, their MMP-9 level was correlated with disease severity, determined by the NYHA functional class. MMP-9 values were also correlated with inflammatory cytokines and neurohormonal factors [11]. The balancing effect for MMP-9 is shown by tissue metalloproteinases inhibitor 1 (TIMP-1), which can also inhibit other MMPs except those associated with the cell membrane. The increased levels of TIMP-1 in tissues and plasma correlate with the degree of myocardial fibrosis and its diastolic dysfunction [12]. TIMP-1 has also been evaluated in studies as a biomarker of myocardial fibrosis [13].

The aim of the study was to assess the occurrence of classic risk factors in the study group of patients with heart failure and to link them with the transcriptional activity of the examined genes: *metalloproteinase 9 (MMP-9)* and the *tissue inhibitor of metalloproteinases 1 (TIMP-1*).

## 2. Materials and Methods

### 2.1. Materials

The study was started after obtaining the consent of the Bioethics Committee of the Medical University of Silesia in Katowice for the research work entitled “Pro- and anti-inflammatory factors and their modulators in patients with various stages of coronary atherosclerosis and its complications” (KNW/0022/KB1/98/I/15/16-24 May 2016) and then continued as part of the project entitled “Transcriptional activity of selected genes in civilization diseases, taking into account the knowledge, lifestyle and pro-health behaviours of the examined patients” (PCN/0022/KB1/36/21-18 May 2021). In accordance with good research practice, written informed consent was obtained from all participants.

The study group consisted of patients consecutively admitted to the Cardiology Department of the Voivodeship Hospital and the Department of Cardiology due to decompensated heart failure in NYHA class III and IV in the course of coronary artery disease, as well as patients with confirmed coronary artery disease, but without signs of heart failure. For the proper assessment of the results of molecular analyses using quantitative RT-qPCR, the study group (*n* = 150; 100%) was divided into patients with coronary artery disease and heart failure (C: *n* = 80; 53.33%), patients without coronary artery disease (A: *n* = 30; 20.00%), and patients with coronary artery disease without heart failure (B: *n* = 40; 26.67%). The control group for the examined patients with heart failure consisted of patients with coronary artery disease excluded in coronary angiography (A) and patients with coronary artery disease without heart failure (B).

The criteria for inclusion in the study group included age over 18 years; expressing written, informed consent; the presence of documented coronary artery disease (significantly and critically stenotic changes and occluded coronary arteries in coronary angiography, condition being present after coronary artery repair surgery or coronary artery bypass surgery and contractility disorders in transthoracic echocardiography); and clinically diagnosed decompensated heart failure in NYHA functional class III-IV.

The criteria for exclusion from the study were lack of consent of the patient to participate, lack of confirmation of coronary artery disease by objective methods, and heart failure caused by other causes, e.g., stroke and mental disorders.

The material for molecular tests was peripheral blood collected from the basilic vein of patients on the 1st day of hospitalisation at the Department of Cardiology.

### 2.2. Methods

The study analysed the transcriptional activity of the genes *metalloproteinase 9 (MMP-9) and tissue inhibitor of metalloproteinases 1 (TIMP-1)* in connection with the presence of risk factors for the development of cardiovascular diseases.

#### 2.2.1. Isolation of Peripheral Blood Mononuclear Cells (PBMCs)

To obtain mononuclear cells, blood was collected in tubes coated with ethylenediaminetetraacetic acid (EDTA) and then centrifuged in a Ficoll gradient to obtain peripheral blood mononuclear cells (PBMCs). The extraction of ribonucleic acid (RNA) from mononuclear cells was performed using TRIzol Reagent (Invitrogen, Waltham, MA, USA), according to a modified method of Chomczyński and Sacchi [14].

#### 2.2.2. Nucleic Acid Extraction

Total RNA extracts were quantified by the spectrophotometric measurement of RNA concentration (Gene Quant II, Pharmacia, NJ, USA). The spectrophotometric evaluation of the extracts included the measurement of absorbance at the wavelengths of 230, 260, 280, and 320 nm and then the determination of the A260/A280 ratio and the protein content. The absorbance value at the wavelength of 260 nm was used to calculate the RNA concentration, assuming that the measurement result in a cuvette with an optical path of 1 cm equal to 1 OD260 corresponds to a concentration of 40 mg of RNA in 1 cm^3^ of the extract.

#### 2.2.3. Real-Time Quantitative Polymerase Chain Reaction (RT-qPCR)

The transcriptional activity of the studied genes was assessed using commercial TaqMan Gene Expression Assay kits labelled at the 5’ ends with the fluorescent dye FAM (6-carboxy-fluorescein) and at the 3’ end with a non-fluorescent quencher. The number of mRNA molecules of the tested genes, MMP-9, TIMP-1, β-actin, and GAPDH, was determined based on the kinetics of the RT-qPCR reaction using the StepOnePlus sequence detector (Applied Biosystems, Foster City, CA, USA) and a kit containing the fluorescent dye ROX TaqMan™ RNA-to-CT ™ 1-Step Kit. RT-qPCR was performed in a single-step reaction mixture containing TaqMan RT-PCR Mix (2×) (AmpliTaq Gold DNA Polymerase), TaqMan RT Enzyme Mix (40×) (ArrayScript™ UP) (Thermo Fisher Scientific, Waltham, MA, USA), TaqMan Gene Expression Assay primers and probe mix (Applied Biosystems, Waltham, MA, USA), a RNA template, and non-pyrogenic water.

Simultaneously with the studied genes, commercially available DNA standards of the β-actin gene were amplified. The reaction mixes for the amplification of β-actin gene DNA standards contained 2× TaqMan^®^ Gene Expression Master Mix (AmpliTaq Gold^®^ DNA Polymerase, dNTP mix, reference ROX), a beta-actin cDNA template, and non-pyrogenic water (Thermo Fisher Scientific, Waltham, MA, USA).

The reverse transcription reaction was carried out as follows: 50 °C for 30 min (reverse transcription), 95 °C for 15 min (polymerase activation), 45 cycles of the two-step reaction (94 °C for 15 s (denaturation)), 60 °C for 1 minute (hybridisation and extension), and the final extension of the amplification products at 72 °C for 10 min. Grounded on the obtained results, a standard curve was prepared for each analysis, based on which the StepOnePlus sequence detector determined the number of copies of the analysed genes. Examined genes transcriptional activity was inferred from the number of mRNA copies per 1 μg of the total RNA.

### 2.3. Statistical Analysis

The obtained results were collected in an Excel spreadsheet and exported to the Statistica 12 software. The mean values, standard deviation (SD), median (Me), and interquartile range (IQR) were calculated. The normality of the distribution of the examined parameters was checked with the Shapiro–Wilk test. The examined parameters did not show the characteristics of normal distribution, so in accordance with the rules of statistical calculations, non-parametric tests were used. To compare the analysed parameters in the study groups, Kruskal–Wallis ANOVA and Mann–Whitney U tests were used. The significance level of p < 0.05 was considered statistically significant.

## 3. Results

The characteristics of the study group participants, taking into account sex and used medications, are presented in Table 1 and Table 2.

Most of the study group participants were men (133; 88.67%), and the average age of the respondents was 65.72 ± 8.95 years.

Over 90% (140; 93.33%) of the respondents used beta-blockers, and over 3/4 (116; 77.33%) of the respondents admitted to using angiotensin-converting enzyme inhibitors.

The characteristics of patients who qualified for the study group, taking into account risk factors for coronary atherosclerosis, are presented in Table 3.

In the study group, the most common classic risk factors were arterial hypertension (117; 78.00%) and overweight/obesity (102; 68%).

The characteristics of the study group participants, taking into account the results of biochemical tests, are presented in Table 4.

The Kruskal–Wallis test showed statistically significant differences between patients with coronary artery disease excluded in coronary angiography (A) and patients with coronary artery disease and heart failure (C) in terms of triglycerides, glucose, and creatinine levels.

The statistical significance (*p*-value) of the difference analysis (Mann–Whitney U test) and descriptive statistics of the transcriptional activity (number of RNA copies) of the *metalloproteinase 9 (MMP-9)* gene family and *tissue inhibitor of metalloproteinases type 1 (TIMP-1)* in patients with coronary angiography excluded coronary artery disease (A), patients with coronary artery disease without heart failure (B), and patients with coronary artery disease and heart failure (C) are presented in Table 5 and Table 6.

The transcriptional activity of the *metalloproteinase 9 (MMP-9)* gene in the group of healthy men was significantly higher compared to women with coronary artery disease excluded in coronary angiography. Moreover, in the group of overweight/obese patients with coronary artery disease and heart failure, a significantly higher transcriptional activity of the *metalloproteinase 9 (MMP-9)* gene was found compared to patients not burdened with this risk factor. The analysis also showed the statistically significant higher transcriptional activity of the *metalloproteinase 9 (MMP-9)* gene in the group of patients with coronary artery disease and heart failure who smoked.

## 4. Discussion

In the study group, the most common classic risk factors were arterial hypertension (117; 78.00%) and overweight/obesity (102; 68%). Epidemiological studies have shown that hypertension is the most important modifiable risk factor in the development of atherosclerosis. Moreover, long-term elevated blood pressure leads to left ventricular hypertrophy and diastolic dysfunction. This condition contributes to increased myocardial stiffness and reduced susceptibility to changes in preload, afterload, and sympathetic tone. Due to the above, antihypertensive treatment should be conducted effectively with the patient’s full involvement in the treatment process, according to the available guidelines in this regard [15]. Despite the above, epidemiological research showed that about 40% of patients with hypertension receive medication, and only one-third of them are successfully treated [16].

Increased body weight, including overweight and obesity, is another important risk factor in the development of cardiovascular diseases based on atherosclerosis. One of the main stages of atherosclerotic plaque formation is endothelial dysfunction and the inflammation of the vascular wall, which is directly related to increased body weight with the accompanying pro-inflammatory and pro-thrombotic conditions in the human body [17]. In addition, there is the phenomenon of lipotoxicity that damages vascular tissues and their functions [18]. Obesity also increases the risk of developing hypertension, diabetes, dyslipidaemia, insulin resistance, metabolic syndrome, and other conditions that are risk factors in the development of atherosclerosis, which was the basis of the disease developed in the studied patients [19]. The presence of obesity together with arterial hypertension and nocturnal hypoxaemia caused by sleep apnoea syndrome, which is common in overweight people, is a predictor of left ventricular hypertrophy, which is another strong risk factor for HF [20,21].

In our study, the Mann–Whitney U test showed a statistically significant difference in the MMP-9 transcriptional activity of the group of patients with coronary artery disease and heart failure (C) who were overweight or obese compared to the other patients from the aforementioned group (*p* = 0.003). Patients with overweight/obesity have higher transcriptional activity than those without the described risk factor. A positive correlation between MMP-9 levels and obesity was confirmed by Unal R. et al., who discovered a positive correlation between MMP-9 and body mass index among 86 nondiabetic patients [22]. Interestingly, in the study by Aksoyer Sezig S. et al., the MMP-9 gene expression was significantly lower in obese patients compared to controls with body weight in normal ranges [23].

A statistically significant difference in the activity of the MMP-9 gene was also observed between smokers and non-smokers with coronary artery disease and heart failure (*p* = 0.035). In their study, Garvin P. et al. showed that MMP-9 levels were correlated with smoking [24]. Also, Nath D. et al., who compared groups of 120 healthy participants, 120 smokers with heart diseases, and 120 active smokers with heart diseases and diabetes, demonstrated that compared to non-smokers, mean serum MMP-9 levels were significantly higher among smokers (*p* < 0.001) [25].

The conducted analyses also revealed a statistically significant difference in the transcriptional activity of the MMP-9 gene in the group of women with coronary artery disease excluded in coronary angiography (A) compared to men from the same group (*p* = 0.009). Tyabjee M. et al. showed, in their research conducted with 204 patients who had coronary artery disease, that MMP-9 protein levels were significantly higher in women than in men [26]. Interestingly, Garvin P. et al. showed that the presence of classic risk factors is not unique in its ability to influence MMP-9 levels. In the mentioned study, MMP-9 levels were associated with psychosocial factors in a middle-aged normal population sample, independent of traditional risk factors. The findings may constitute a possible link between psychosocial factors and cardiovascular risk, which can also be a starting point for research in this area [27].

Taking into account the risk factors, no significant differences were found between the study groups in the expression of the *tissue inhibitor of metalloproteinases 1 (TIMP-1)* genes; this is not consistent with the available literature, because the results of the study by Sundström J. et al. indicated a direct relationship between the levels of TIMP-1 in plasma with all the classic risk factors for cardiovascular disease. These researchers also showed that plasma TIMP-1 levels were elevated in diabetic patients on antihypertensive treatment [28]. Moreover, TIMP-1 gene transcriptional activity correlated negatively with the advancement of heart failure (decrease in left ventricular ejection fraction) [29].

Despite significant advances in risk prediction, cardiovascular diseases remain the leading cause of death worldwide, and multifaceted, non-pharmacological lifestyle interventions aimed at value awareness through the control of modifiable risk factors reduce the risk of fatal cardiovascular events. In addition, these interventions reduce the number of readmissions associated with heart failure decompensation compared with usual care. Therefore, they should be included in secondary prevention programs aimed at patients with cardiovascular diseases, including HF, due to their beneficial effect on the mortality and morbidity of patients in this group. The best confirmation of the impact of reducing the number of risk factors on long-term treatment outcomes is the data from the study presented below. Among the 20,900 physicians who participated in the Physicians’ Health Study conducted at Harvard University in the United States, it was proven that the lifetime risk of heart failure was higher in those with hypertension than in those without hypertension and that healthy habits associated with lifestyle factors such as maintaining a healthy body weight, not smoking, engaging in regular physical activity, and eating fruit and vegetables regularly were, both individually and collectively, associated with a lower lifetime risk of developing heart failure. The highest risk of developing the discussed disease was found in respondents who did not adhere to any of the analysed components of a healthy lifestyle (21.2%; 95% CI: 16.8%–25.6%), and the lowest risk was found in those who observed at least four (10.1%; 95%%PU: 7.9%–12.3%) [30].

Study limitations include, but are not limited to, the following: the small number of respondents, the small number of women participating in the study, and taking into account only a few risk factors of cardiovascular disease during analysis. Moreover, not all patients have both coronary angiography and CCT to avoid exposing them to overly high doses of radiation. CCT was additionally performed only in the patients for whom the above-mentioned doctors did not agree on the final result of coronary angiography. It is worth noting that the observed higher transcriptional activity of the MMP-9 gene was associated with the presence of the most common risk factors. It cannot be ruled out that if the study group was enlarged, the observed effect (increased transcriptional activity of the MMP-9 gene) would also be visible in connection with the other mentioned factors. Despite this, the conducted studies provide direct evidence linking the presence of risk factors with the molecular activity of some proatherogenic genes and, consequently, the development of cardiovascular diseases. Further research should include deepening the existing knowledge in this research area.

## 5. Conclusions

The examined patients with heart failure due to myocardial ischemia were burdened with numerous cardiovascular risk factors, the most common of which were arterial hypertension, obesity/overweight, and hypercholesterolemia. A significant increase in the transcriptional activity of the *metalloproteinase 9 (MMP-9)* gene in the presence of risk factors (male sex, overweight/obesity, and smoking) indicates another pathomechanism of their action and participation in the development and progression of heart failure in the course of myocardial ischemia. There is a need for systematic information and educational activities promoting a healthy lifestyle with the elimination of modifiable risk factors for cardiovascular diseases.

## Figures and Tables

**Table 1 biomedicines-12-00601-t001:** Characteristics of the study group taking into account sex and age.

Variables	Sex				Age [Years]	
	Women		Men		X ± SD	Me
Groups	*n*	%	*n*	%		IQR
**ABC**	17	11.33	133	88.67	65.72 ± 8.95	65.00 (13.00)
**(150; 100%)**						
**A**	15	50	15	50	62.77 ± 9.29	64.50 (11.00)
**(30; 20.00%)**						
**B**	1	2.5	39	97.5	64.67 ± 10.45	63.00 (18.00)
**(40; 26.67%)**						
**C**	1	1.25	79	98.75	67.35 ± 7.67	66.00 (11.50)
**(80; 53.33%)**						

**Explanation of abbreviations:** A—patients with coronary artery disease excluded in coronary angiography; B—patients with coronary artery disease without heart failure; C—patients with coronary artery disease and heart failure; *n*—number of participants.

**Table 2 biomedicines-12-00601-t002:** Characteristics of the study group taking into account used medications.

Variables	Study Group (150; 100%)							
	ABC		A		B		C	
	*n* = 150 (100%)		*n* = 30 (20%)		*n* = 40 (26.67%)		*n* = 80 (53.33%)	
	*n*	%	*n*	%	*n*	%	*n*	%
**Number (*n*, %) of the study group**	150	100	30	20	40	26.67	80	53.33
**Number (*n*, % of a given group)**	150	100	30	100	40	100	80	100
**ACEI**	116	77.33	14	46.66	38	95	64	80
**ARB**	8	5.33	8	26.67	0	0	0	0
**Statins**	134	89.33	18	60	40	100	76	95
**BB**	140	93.33	22	73.33	40	100	78	97.5
**ASA**	111	74	8	26.66	37	92.5	66	82.5

**Explanation of abbreviations:** A—patients with coronary artery disease excluded by coronary angiography; B—patients with coronary artery disease without heart failure; C—patients with coronary artery disease and heart failure; *n*—number of participants; ACEI—angiotensin-converting enzyme inhibitors; ARB—sartans; BB—blockers of beta receptors; ASA—acetylsalicylic acid.

**Table 3 biomedicines-12-00601-t003:** Characteristics of the study group including some classic risk factors for coronary atherosclerosis.

Variables	Study Group (150; 100%)							
	ABC		A		B		C	
	*n* = 150 (100%)		*n* = 30 (20%)		*n* = 40 (26.67%)		*n* = 80 (53.33%)	
	*n*	%	*n*	%	*n*	%	*n*	%
**Sample size (*n*, % of the study group)**	150	100	30	20	40	26.67	80	53.33
**Sample size (*n*, % of a given group)**	150	100	30	100	40	100	80	100
**Positive family history of cardiovascular diseases**	9	6.00	5	16.66	2	5	2	2.50
**Hypertension**	117	78.00	19	63.33	40	100	58	72.50
**Diabetes**	52	34.66	6	20.00	12	30	34	42.50
**Hypercholesterolaemia**	78	52.00	7	23.33	21	52.50	50	62.50
**Overweight/obesity**	102	68.00	14	46.66	33	82.50	55	68.75
**Tobacco smoking**	47	31.33	3	10.00	11	27.50	33	41.25

**Explanation of abbreviations:** A—patients with coronary artery disease excluded in coronary angiography; B—patients with coronary artery disease without heart failure; C—patients with coronary artery disease and heart failure; *n*—number of participants.

**Table 4 biomedicines-12-00601-t004:** Characteristics of the study group, taking into account the results of biochemical tests.

Variable	Study Group (150; 100%)								*p*
	ABC		A		B		C		
	*n* = 150 (100%)		*n* = 30 (20%)		*n* = 40 (26.67%)		*n* = 80 (53.33%)		
	X	Me	X	Me	X	Me	X	Me	
	±SD	IQR	±SD	IQR	±SD	IQR	±SD	IQR	
**Total cholesterol [mg/dL] (N: 115–190)**	157.30	155.00	160.70	161.50	154.92	151.00	157.21	154.50	NS
	±38.51	52.00	±30.25	26.00	±36.40	47.50	±42.43	62.00	
**HDL cholesterol [mg/dL]**	44.56	43.25	46.63	45.00	46.76	45.10	42.68	40.85	NS
**(Women N: >45)**	±11.47	15.50	±10.14	12.00	±9.44	13.30	±12.59	18.55	
**(Men N: >40)**									
**LDL cholesterol [mg/dL]**	87.01	83.80	87.06	87.50	78.73	76.00	91.13	85.50	NS
	±32.90	43.80	±21.32	19.00	±31.76	43.90	±36.39	50.45	
**Triglycerides**	133.89	120.00	146.76	150.00	153.50	145.00	119.26	92.50	*p* < 0.001
**[mg/dl] (N: <150)**	±63.63	81.00	±41.55	40.00	±64.37	81.50	±66.94	68.50	
**Glucose concentration in blood serum [mg/dL]**	113.43	104.40	102.06	64.00	107.01	99.90	120.89	114.30	*p* < 0.001
**(N: 74–106)**	±26.80	30.60	±20.66	12.00	±22.35	9.00	±28.72	33.30	
**Creatinine [mg/dL] (N:0.67–1.17)**	1.20	1.09	0.94	0.91	1.00	1.03	1.40	1.26	*p* < 0.001
	±0.52	0.40	±0.25	0.32	±0.21	0.22	±0.61	0.48	

**Explanation of abbreviations:** A—patients with coronary artery disease excluded in coronary angiography; B—patients with coronary artery disease without heart failure; C—patients with coronary artery disease and heart failure; N—diagnostic range; *n*—number of participants; X—mean; ±SD—deviation standard; Me—median; IQR—interquartile range (Kruskal–Wallis test).

**Table 5 biomedicines-12-00601-t005:** Transcriptional activity of *metalloproteinase 9* and *tissue inhibitor of metalloproteinases type 1* in the study group, taking into account selected risk factors.

Risk Factors		MMP-9	TIMP-1
**Sex**	women A vs. men A	0.009 *	0.775
	women B vs. men B	---------	---------
	women C vs. men C	---------	--------
**Arterial hypertension**	A without arterial hypertension vs. A with arterial hypertension	0.200	0.134
	B without arterial hypertension vs. B with arterial hypertension	----------	---------
	C without arterial hypertension vs. C with arterial hypertension	0.525	0.659
**Diabetes**	A without diabetes vs. A with diabetes	0.631	0.899
	B without diabetes vs. B with diabetes	0.906	0.119
	C without diabetes vs. C with diabetes	0.691	0.619
**Overweight/** **Obesity**	A with overweight/obesity vs. A without overweight/obesity	0.637	0.093
	B with overweight/obesity vs. B without overweight/obesity	0.545	0.161
	C with overweight/obesity vs. C without overweight/obesity	0.003 *	0.497
**Tobacco smoking**	A non-smoker vs. A smoker	0.457	0.086
	B non-smoker vs. B smoker	0.388	0.575
	C non-smoker vs. C smoker	0.035 *	0.075
**Lipid disorders**	A without lipid disorders vs. A with lipid disorders	0.104	0.065
	B without lipid disorders vs. B with lipid disorders	0.894	0.226
	C without lipid disorders vs. C with lipid disorders	0.292	0.598

**Explanation of abbreviations:** A—patients with coronary artery disease excluded in coronary angiography; B—patients with coronary artery disease without heart failure; C—patients with coronary artery disease and heart failure; MMP-9—metalloproteinase 9; TIMP-1—metalloproteinase inhibitor 1; *—statistically significant result (Mann–Whitney U test).

**Table 6 biomedicines-12-00601-t006:** Descriptive statistics of the transcriptional activity (number of mRNA copies) of the MMP-9 gene, taking into account the risk factors of cardiovascular diseases that significantly influence it.

Analysed Group	Risk Factor	Variables	M	SD	Me	Q1	Q3	Min.	Max.
**A**	**Sex**	**Males**	23,258.6	179,057	22,974.8	2160.5	13,289.4	115.9	69,889.1
		**Females**	11,338.9	19,839.3	5063.6	7306.8	33,176.4	851	70,738.7
**C**	**Overweight/** **Obesity**	**Yes**	15,755.9	10,440.4	9967.4	4347.4	16,716	1277.3	36,081.6
		**No**	12,844.8	28,455.1	6693.4	1772	15,039.7	83	175,052.2
	**Tobacco smoking**	**Yes**	18,661.1	31,285.2	9457.1	2122.6	19,227.6	250.4	175,053
		**No**	9941.7	9607.6	6693.4	2530.6	12,538.5	82.9	36,199.8

**Explanation of abbreviations:** A—patients with coronary artery disease excluded in coronary angiography; C—patients with coronary artery disease and heart failure; M—median; Me—median; Min.—minimal value; Max.—maximal value; Q1—lower quartile; Q3—upper quartile; SD—standard deviation.

## Data Availability

All datasets are available from the corresponding author upon reasonable request.

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
