# Peer review of "Risk Factors Association with Transcriptional Activity of Metalloproteinase 9 (MMP-9) and Tissue Inhibitor of Metalloproteinases 1 (TIMP-1) Genes in Patients with Heart Failure"

_biomedicines, 2024, doi:10.3390/biomedicines12030601_

Round 1

Reviewer 1 Report (Previous Reviewer 4)

Comments and Suggestions for Authors

This paper was well revised and this reviewer has no further comment.

Author Response

Response to the Reviewer comments regarding article entitled: 

“Risk factors association with transcriptional activity of metalloproteinase 9 (MMP-9) and tissue inhibitor of metalloproteinase 1 (TIMP-1) genes in patients with heart failure” (Biomedicines-2731185)

“This paper was well revised and this reviewer has no further comment.”

Dear Reviewer,

 Thank you once again for all your valuable comments and suggestions.

 Your sincerelly,

 Józefa Dąbek

Reviewer 2 Report (Previous Reviewer 5)

Comments and Suggestions for Authors

Author, Józefa Dąbek, is to be congratulated for a nicely presented revised manuscript and the effort.

The authors have made significant strides in addressing my concerns in this version, although a few remaining issues still need attention.

Furthermore, I have a suggestion for improvement.

In your responses to reviewers, please provide the page and line numbers of the revised sections, as this greatly enhances the efficiency of the review process.

While the Introduction and Discussion sections have been revised, they remain lengthy and challenging to navigate.

For instance, the section on Page 2, Lines 56-67, which discusses rare causes of ischemic heart disease in the Introduction, could be more concise.

In the Discussion section, the crux of the paper's key points emerges around Page 11, Line 380, and the preceding text is overly verbose.

While investigating MMP-9 gene activity for personalized medicine holds significance, the discussion on its link to lifestyle habits appears somewhat tangential.

The first half of the Discussion dwells extensively on this tangent, which could be streamlined for clarity.

I recommend providing a more succinct description instead.

Author Response

Dear Reviewer,

I attach responses to you comments in word file.

Your sincerely,

Józefa Dąbek

This manuscript is a resubmission of an earlier submission. The following is a list of the peer review reports and author responses from that submission.

Round 1

Reviewer 1 Report

Comments and Suggestions for Authors

The manuscript entitled “Risk factors association with transcriptional activity of metallo-2 proteinase 9 (MMP-9) and tissue inhibitor of metalloproteinase 3 1 (TIMP-1) genes in patients with heart failure” by Dąbek et al, aims to evaluate the appearance of classic risk factors in the study group of patients with heart failure and link them with the transcriptional activity of the genes examined: metalloproteinase-9 (MMP-9) and tissue inhibitor of metalloproteinases 1. (TIMP-1). The work could be interesting, but it has some very important aspects to consider:

- In the introduction, the authors provide a general description of heart failure. However, the reason why they will study metalloproteinase-9 (MMP-9) and tissue inhibitor of metalloproteinases 1 (TIMP-1) is not described. It is important that the hypothesis and objectives of the work have a rational basis.

- Why did they not use a group of healthy patients as a control? This must be justified.

- Table 3 is extremely confusing. This must be completely re-elaborated for a better understanding of the reader.

- How can you obtain conclusions from groups of women B (patients with coronary artery disease without heart failure) and C (patients with coronary artery disease and heart failure) when n is equal to 1? The authors must justify this observation.

-             Discussion:

- It is very extensive, speculative and at no time does it conclusively address the importance of determining the parameters studied.

- How much do the MMP-9 and TIMP-1 values obtained in the studied groups (A, B and C) differ from the values of healthy individuals? This point is important to know what value this study may have.

- In the discussion the authors define renal dysfunction having determined only plasmatic creatinine. To define this entity, other determinations must be made. This point must be corrected.

Reviewer 2 Report

Comments and Suggestions for Authors

The authors evaluated the presence of traditional risk factors in a group of 21 patients with heart failure and established connections between these factors and the transcriptional activity of 22 specific genes, namely, metalloproteinase-9 (MMP-9) and tissue inhibitor of metalloproteinases 1 (TIMP-1).

The following comments and suggestions may enhance the insights of the study:

1- The introduction lacks information about MMP-9 and TIMP-1, and why the authors will use them as indicators.

2- The authors enumerate six risk factors in table 4 associated with the transcriptional activity of MMP-9 and TIMP-1. They discuss only the correlation of obesity with MMP-9 and TIMP-1, please add the insights of your results in Table 4 with Sex, Arterial Hypertension, Diabetes, Smoking, and Lipid Disorders. This is the main title of your manuscript.

3- Figure 1, add lines to the x, and y axes.

4- Regarding sex as a risk factor, I think the number of women who participated in the study (11%) was not enough regarding the sample size used. In addition, I think you can add it as a limitation of your study.

5- The mean age is 65 years old, and your inclusion criteria started with 18 years old, Please add information about the age of the 150 participants.

6- The authors focused on previous studies in their discussion, they can correlate them with the is results.

7- The paragraph started at line 331, can be moved to be merged at line 307, to reflect obesity as a risk factor.

Comments on the Quality of English Language

The English style is fine

Reviewer 3 Report

Comments and Suggestions for Authors

This article entitled “Risk factors association with transcriptional activity of metalloproteinase 9 (MMP-9) and tissue inhibitor of metalloproteinase 1 (TIMP-1) genes in patients with heart failure” by Dabek et al. reports 1) distribution of risk factors for heart diseases such as hypertension, 2) blood biochemical levels such as glucose, and also 3) MMP9 and TIMP1 expression levels in the blood samples of total 150 patients. While there is no doubt that cardiovascular diseases are top causes of death, therefore the topic is of interest to many, the significance of this particular study is not very clear especially because the effects of the diseases on MMP9/TIMP1 expression levels have been previously reported (of which the authors failed to replicate for TIMP1 due to the small sample size). There are other concerns listed below:

Major Points

It is not clear how “patient group A” was determined. Without coronary angiography, how do you define coronary artery disease? And how does that affect to the distributions listed in Table 2?

Table 4 seems to report only p values (although not defined anywhere). In Abstract and Conclusion, the authors repeatedly describe “higher transcriptional activity” without showing any numbers.

Minor Points

L254-L255: Is this statement based on reference [12]? If so, this reference does not mention it. If not, what is it based on? Because the difference is not that big according to US dep of health and human services (https://www.womenshealth.gov/node/1374#:~:text=%231%20%E2%80%93%20Heart%20Disease%20is%20the,some%20form%20of%20heart%20disease.)

Anything to say about why MMP9/TIMP1 may increase in cardiovascular disease patients? Is it the cause or the result of compensation? Why certain factors such as sex showed difference but not others such as hypertension? Can you relate to functions of MMP9/TIMP1?

Reviewer 4 Report

Comments and Suggestions for Authors

This clinical study aimed to evaluate the prevalence of classic risk factors in patients with heart failure and their association with the transcriptional activity of specific genes: metalloproteinase-9 (MMP-9) and tissue inhibitor of metalloproteinases 1 (TIMP-1). The authors demonstrated a significant increase in MMP-9 gene transcriptional activity in the presence of risk factors such as male gender, overweight/obesity, and smoking. This finding suggests an alternative pathomechanism of these factors in the development and progression of heart failure during myocardial ischemia

This reviewer who investigated serum MMP-9 concentrations in heart failure patients 20 years ago without yielding significant results and found this article particularly intriguing. There are several comments as described below. 

Major comments:

1.       The authors need to include demographic data in the baseline characteristics of the patients, including age, gender, and medications, particularly statins. 

2.       The absence of Table 1 in the document raises questions. Could the authors clarify this?   

3.       This study should explore which risk factors correlate with heart failure in the transcriptional activity of the MMP-9 gene, possibly using multivariate analysis.

4.       The influence of systemic organs, apart from the heart, on the transcriptional activity of the MMP-9 gene, as assessed through blood samples, should be addressed. How do the authors account for this factor?

5.       The discussion section is overly lengthy and requires condensation. The authors should concentrate more on elucidating the mechanisms of MMP-9 and TIMP-1 in heart failure.

Reviewer 5 Report

Comments and Suggestions for Authors

In the study by Józefa Dąbek, they investigated “Risk factors association with transcriptional activity of metallo-2 proteinase 9 (MMP-9) and tissue inhibitor of metalloproteinase 3 1 (TIMP-1) genes in patients with heart failure”.

This is a very interesting study that investigates MMP-9 and TIMP-1, which are attracting attention as promoting factors of arteriosclerosis, in clinical practice. However, the following concerns need to be addressed.

Concern #1 

Your introduction seems a bit long and contains unnecessary sentences. Your research was investigating how TIMP-1 and MMP‐9 behave in clinical practice of heart failure treatment.

TIMP-1 and MMP-9 can be involved in the progression of arteriosclerosis. Since it was reported TIMP-1 inhibits MMP, there has been confusion as to whether they are acting as an accelerator or a brake against arteriosclerosis. This important issue is not mentioned at all in the introduction. I recommend adding information about this issue and reducing epidemiological information about heart failure in introduction.

Concern #2

The average age of the subjects in this study was relatively high at 65 years old. Could it be guaranteed that there was no arteriosclerosis in this group based solely on coronary angiography? Was coronary CT used in the investigation of arteriosclerosis? Coronary angiography alone cannot determine balanced positive and negative remodeling of arteriosclerosis. 

CCT has been the gold standard method for evaluating arteriosclerosis without considering the influence of remodeling.

If CCT was not performed, it is a significant limitation of this study.

Concern#3

TIMP-1 is an inhibitor of MMPs, theoretically, MMP-9 and TIMP should influence each other. How do you explain that only MMP-9 showed characteristic dynamics with gender, obesity, and smoking? Does something else affect the relationship between TIMP-1 and MMP-9 ?

Concerb#4

Your discussion was also too long. The point of your article was that men, obesity, and smoking, which has been well-known risk factors for atherosclerosis, were associated with the changing of MMP-9 in patients with heart failure and atherosclerosis. This was an very interesting outcomes, however the impact of heart failure and MMP-9 was not discussed at all. It has been a well-known fact that severe ischemia has not been necessarily cause heart failure, but this relation may be involved in the onset of heart failure. Please discuss this issue.

Concern#5

Furthermore, the epidemiological information overlapped considerably with the introduction. Please limit such information to a minimum. Important to this article are the literature review of past studies that investigated the impact of MMP-9 and TIMP-1 on atherosclerosis and/or heart failure, and the insights added by your research. These perspectives are more important in the discussion.